# The Power of We

**DOI:** 10.3390/v15040921

**Published:** 2023-04-05

**Authors:** James W. Le Duc

**Affiliations:** Galveston National Laboratory, University of Texas Medical Branch, Galveston, TX 77555, USA; jwleduc@utmb.edu

**Keywords:** hantaviruses, diagnosis, epidemiology, ecology, transmission, Seoul virus, Puumala virus, hemorrhagic fever with renal syndrome

## Abstract

“The Power of We” is a personal tribute to the individuals and organizations that collaborated in the discovery and advancement of knowledge of the hantaviruses following the original isolation of Hantaan virus by Ho Wang Lee. It focuses on the work done primarily at the United States Army Medical Research Institute of Infectious Diseases during the decade of the 1980s under the leadership of Joel Dalrymple, who worked in close partnership with Ho Wang Lee. These early studies helped define the global distribution of Seoul virus and provided seminal information on its maintenance and transmission among urban rats. Other collaborations involved partners in Europe, Asia, and Latin America and resulted in the isolation of novel hantaviruses, a better understanding of their global distribution, and validation of diagnostics and therapeutic interventions for treatment of human diseases. By working in partnership, scientists from around the world made critical discoveries that led to a better understanding of the hantaviruses. “The Power of We” demonstrates that we all benefit when we work together with a shared vision, a common commitment to excellence, and mutual respect.

## 1. Introduction

I was fortunate to be recognized with the Ho Wang Lee Award and invited to offer a keynote address during the Xth International Conference on HFRS, HPS and Hantaviruses held in Fort Collins, Colorado in 2016. This chapter is drawn from that presentation and represents my sincere tribute to the many friends and colleagues that I had the honor of working with during the early days of discovery following the isolation of Hantaan virus by Ho Wang Lee [1]. It is not meant to be a comprehensive review of all the exciting accomplishments made. Rather, it is a personal account of some of the fascinating work done and a tribute to the accomplished individuals with whom we collaborated and the incredible discoveries that were made—thus, “The Power of We”.

With the availability of Hantaan virus, nearly unlimited opportunities emerged for research on the clinical characteristics of the human disease, exploration of therapeutics, development of diagnostics, molecular and taxonomic characterization of the virus, and detailed investigations of the epidemiology and ecology of the new pathogen. Small groups of investigators formed the early “We”, including those who had been studying hemorrhagic fever with renal syndrome (HFRS) even before the causative agent was identified. These included Carlton Gajdusek’s group at the US National Institutes of Health and scientists in Russia, Scandinavia, the former Yugoslavia, and, of course, Asia, especially in China, Korea, and Japan. With the isolation of Hantaan virus, “We” grew, and like the branches of a tree, collaborations formed limbs of “We” that included partners from around the world. This is the story of one branch, those of us working at USAMRIID and the many collaborations that developed during the exciting decade of the 1980s. In some respects, we were fortunate to have a “dream team” of wonderfully talented and inquisitive minds that formed a strong bond with the common goal of understanding this newly discovered virus—all a result of the persistence and commitment of Ho Wang Lee, who had finally discovered the cause of this devastating disease.

The USAMRIID program was led by Joel Dalrymple. Joel had developed a strong friendship with Ho Wang Lee, who had been supported by US Army grants for many years to investigate the cause of Korean hemorrhagic fever (KHF), which was first recognized by Western medicine during the Korean War. The USAMRIID “We” was born through the collaboration between Ho Wang and Joel, a partnership that lasted until Joel’s untimely passing in 1992, grew to include many leaders in the field, and produced a remarkable set of seminal discoveries that have benefited the world. Very early on, “We” grew when Joel recruited Connie Schmaljohn to join the team, and she quickly established herself as an exceptional leader as she and Joel pioneered the molecular characteristics of the new virus.

## 2. Diagnosis of Hantaviruses

The US military’s interest in the hantaviruses stemmed from the Korean conflict, when troops suffered from a disease unknown to Western medicine, which was named KHF. Some of the best minds in medicine at that time helped define the clinical course of the disease but were unable to identify the etiologic agent. Among their many efforts was one to collect acute and convalescent sera from suspected cases seen between 1951 and 1954. These sera were lyophilized in glass vials, sealed by melting the tops of the glass containers, and labeled with adhesive tape that included the typed patient information, including the day of collection post onset of clinical symptoms. These were then stored in three olive drab military foot lockers until we opened them for testing. This collection proved incredibly valuable when Hantaan virus was finally isolated and testing for antibodies among these known patients became possible. The sera were rehydrated and tested for IgM and IgG antibodies using an enzyme immunoassay technique adapted for hantaviruses by Tom Ksiazek and Jim Meegan. Virtually all patients clinically diagnosed with KHF were IgM positive by day 5 of the illness, and all subsequently developed a robust IgG response. This study provided both validation of an important diagnostic tooling and definitive confirmation that KHF of the Korean conflict was due to Hantaan virus infection [2] (Table 1).

The enzyme immunoassay was not, however, the first attempt to develop a diagnostic assay. Earlier, George French used Hantaan virus-infected cells dried on spot slides for immunofluorescent immunoassays (IFA) to detect antibodies to hantaviruses. This technology was essential to the early discovery of hantaviruses around the world, but the slides were difficult to produce, store, and transport, and results were sometimes challenging to interpret due to non-specific reactions. Nonetheless, this early diagnostic tool was critical to the unfolding story of the hantaviruses.

## 3. Hantaviruses in Domestic Rats

Shortly after the isolation of Hantaan virus from the Korean field mouse, *Apodemus agrarius*, Ho Wang and his colleagues discovered a Hantaan-like virus found in urban rats in Seoul, Korea. The virus was associated with a human disease similar to KHF and raised concern that perhaps Hantaan virus had jumped hosts from the rural *Apodemus* rodents to urban rats [3]. Domestic rats are ubiquitous in cities around the world, and the fear that rats might harbor Hantaan virus caused us to explore if rats in US cities, and especially those near port cities receiving international ships, might be infected with Hantaan virus. As a result, we set out to capture rats in several US port cities and were concerned to find that many were infected with a hantavirus (Table 2) [4]. Initial studies relied on IFA serological tests of rat sera, but we were also able to isolate the virus from wild-caught rat tissues. The viruses we isolated from rats caught at US ports were nearly identical to those found by Ho Wang and his colleagues in Korea, and these rat-associated viruses turned out to be a close relative of Hantaan virus, but distinct. The new virus was named Seoul virus after the site of its first isolation.

At about the same time, we became aware of laboratory workers suffering from acute renal failure at a university in Belgium, and we were able to trace back the technician’s exposure to a specific inbred strain of laboratory rats. Working with Karl Johnson, who was then at USAMRIID, and Jan Desmyter and his colleagues from Belgium, we were able to show that the lab rats harbored a Seoul-like virus that was the likely source of the infections [5]. Laboratory rats infected with hantaviruses were subsequently found in other laboratory animal handling facilities, especially in Asia, and pet rats were later shown to be a source of human infection in many locations in the USA and Europe as well [6].

We began what turned out to be a long-term collaboration with Jamie Childs and his team at Johns Hopkins University in Baltimore, MD in the early 1980s. Johns Hopkins has a wonderful history in public health, including past leaders who had studied urban rats in detail; consequently, we inquired if any studies were underway that might be leveraged to include the study of the newly discovered hantaviruses of domestic rats. Fortuitously, Jamie Childs was leading such studies along with a small team of experts that included Greg Glass and George Korch (who later became USAMRIID Commander). They welcomed the invitation to collaborate and immediately recognized the tremendous opportunity to help define the ecology of this new pathogen.

Over the course of the next several years, Jamie and his team intensively investigated the prevalence of hantaviruses in Baltimore rats. They discovered that hantaviruses were especially prevalent in the rats, with antibody prevalence rates increasing as rats aged. About a third of newborn rats had maternal antibodies to the virus that waned over time as the rats matured, then increased as they reached adulthood. The oldest adult rats as estimated by body weight had antibody prevalence rates nearing 90% [7].

Jamie and his team also conducted an incredible study to determine the incidence of hantavirus infection among free-living rats. They used a mark-release-recapture study design in several Baltimore neighborhoods that ran between 1984 and 1986. Rats were captured and anesthetized, their blood was drawn, and they were individually tagged and then released back at the exact spot where they were captured. Traps were set at each site monthly, and recaptured individual rats were again bled and then released. Depending upon the location sampled, the percentage of rats that seroconverted from antibody negative to positive ranged from about 17% to 50%, indicating that the virus was actively transmitted among rats in many Baltimore neighborhoods [8].

Given the significant level of hantavirus infection among Baltimore rats, it was obvious that we should look for associated human disease. Jamie and his team obtained over 1800 sera from two sources in Baltimore and examined them for antibodies to hantaviruses. From this sample, four individuals had significant antibody titers identified by IFA and enzyme immunoassay (EIA) to the Seoul virus isolated from Baltimore rats. These results were confirmed by plaque reduction neutralization tests. While no clinical record of acute illness was available, these serological results clearly established that humans were infected with the virus [9,10]. Subsequent studies suggested that seropositive individuals may have been more likely to suffer from chronic kidney disease, but attempts to follow up on this observation have yielded mixed results [11].

## 4. How Are Hantaviruses Transmitted?

We questioned how the virus was being transmitted and approached this through two separate studies. Using the unique aerosol-generating capabilities found at USAMRIID, Ed Nuzum led a team to study the sensitivity of laboratory rats to aerosol exposure of either Hantaan, Seoul, or Puumala virus. Various concentrations of each virus were aerosolized and exposed to small groups of Wistar rats. A second group of rats received the same viruses at similar concentrations but administered by intramuscular inoculation. All rats were tested for antibodies to each virus at 28–30 days post exposure, and the percentage of rats seroconverting was found to increase with increasing virus concentrations until all animals were positive at the highest dose administered. Surprisingly, we found that those exposed by intramuscular inoculation were infected with a much smaller dose of the virus as compared to those exposed by aerosol [12].

The finding that rats were more easily infected by intramuscular inoculation as compared to aerosol exposure suggested to us that perhaps the virus may have been more frequently transmitted by bite among rats. To test this second hypothesis, Greg Glass reviewed rates of wounding among the captured free-living rats. Using weight-based age estimates, we found that the rates and severity of wounds increased with age, especially among males, and that this increase in wounding paralleled the increase in antibody prevalence [13]. Transmission of hantaviruses by bite from rodents to humans had been documented, and the virus was known to be present in rodent saliva, so territorial fighting among male rats and wounding during mating could represent an important route of transmission in urban rats.

The Baltimore rat studies led by Jamie Childs and his team were incredibly detailed and still today represent perhaps the most definitive description of the maintenance of a hantavirus in nature [14].

## 5. Hantaviruses in Latin America

Realizing that urban rats are ubiquitous, we built on our connections with collaborators in Latin America to explore if rat-associated hantaviruses were equally prevalent there. The US Army had maintained a small research laboratory in conjunction with the Institute Evandro Chagas in Belem, Brazil, at the mouth of the Amazon during much of the 1970s, and I was fortunate to serve as the Army Laboratory Commander. Unfortunately, we were forced to close the laboratory due to political challenges in the late 1970s, and as the last Commander, I was responsible for disbanding the program and returning the small group of civilian and military Americans to the States. Fortunately, the scientific partnerships that we had established remained strong, and our colleagues in Latin America were eager to rekindle our work together to address this novel virus. As we were collecting rats from port cities in the USA, we also worked with our partners in Brazil and Argentina, and between September 1982 and March 1983, we captured and tested rats for evidence of hantavirus antibodies and attempted virus isolation from wild-caught rat tissues. We found that more than half the rats captured in Belem were antibody positive, while 6–14% of rats from Sao Paulo and Recife were also positive, and more than 10% of rats from Buenos Aires were likewise positive. Finally, we were able to isolate the virus from the lung tissues of rats captured in Belem (Table 3) [15].

Given that we had been successful in finding hantavirus-infected rats in many locations in both North and South America, we expanded testing to include samples provided by many partners from around the world. We engaged many of our friends and colleagues from around the world in an attempt to determine the global distribution of hantaviruses in peridomestic rodents. Over 1700 sera from more than 20 sites worldwide were tested for anti-hantavirus antibodies. Antibody-positive rodents were found at almost every location sampled, with domestic rats (*Rattus norvegicus*) having the highest prevalence [16]. Clearly, the rodent-associated hantaviruses were widely distributed and had been for many years. Our discoveries reflected the new tools we had to detect the virus rather than a recent dissemination of the virus.

## 6. Ribavirin Trial in China

Recognizing the importance of Hantaan virus as a cause of severe human illness in Asia, USAMRIID began an effort to determine if ribavirin, an antiviral drug known to be efficacious in treating other hemorrhagic fevers, might also be effective in treating HFRS. John Huggins led efforts to establish a collaboration with scientists and physicians in Wuhan, China, to conduct a clinical trial of ribavirin in the treatment of epidemic hemorrhagic fever, the name used in China for Hantaan virus infections. The goal was to complete a double-blind, placebo-controlled trial of ribavirin in hospitalized patients, and the challenges were enormous. In the mid-1980s, China had just emerged from the Cultural Revolution and was still suffering from its aftermath. This was still the “old China”, where most people used bicycles to get around and lived very meagre lifestyles. The fact that we were able to establish a formal collaboration between USAMRIID and the Chinese government was itself amazing. There was a need for modern laboratory instrumentation to be available in Wuhan to conduct the essential laboratory tests needed to confirm patient diagnoses and to monitor clinical parameters. This was especially difficult given the lack of modern clinical laboratory capacity in Wuhan. Consequently, we imported new laboratory instrumentation and set it up in Wuhan. Diagnostic testing for hantavirus infection was established, and a young physician-scientist named Shu-Yuan Xiao was trained by Jim Meegan to conduct the IgM diagnostic test to be used to confirm the clinical diagnosis of enrolled patients.

Between 1985 and 1987, we conducted a prospective, randomized, double-blind, placebo-controlled trial of the efficacy of intravenous ribavirin in 242 serologically confirmed HFRS patients. When the code was broken after three transmission seasons, we found that survival was significantly enhanced, and disease severity was reduced among ribavirin-treated patients [17]. This study was noteworthy in several ways. It demonstrated the immense value of global collaborations to systematically evaluate promising clinical interventions regardless of location or existing capacity. And it clearly offered a new diagnostic capability and a novel treatment option for a serious endemic disease in China, where heretofore neither had existed. Perhaps most importantly, it fostered further scientific collaborations between US and Chinese scientists. The successful drug trial opened the door for a few Chinese scientists to join USAMRIID through a fellowship program that allowed them to work side by side with US scientists on the hantaviruses and other topics. These partnerships led to lifelong friendships that continue to this day, friendships that have significantly benefited both the USA and China.

## 7. Puumala Virus in Sweden

A form of HFRS was known in Scandinavia and much of Europe prior to the isolation of Hantaan virus by H.W. Lee; however, the causative virus had not been isolated, and the detailed ecology and epidemiology of the disease in Sweden was lacking. In Scandinavia, the human disease was known as nephropathia epidemica and was caused by Puumala virus, a close relative to the prototype Hantaan virus. Our studies of a hantavirus in Sweden began when I was fortunate to meet a young Swedish physician, Bo Niklasson, at a meeting in the early 1980s, and we discovered our common interest in the hantaviruses. This led to a series of joint field trips and laboratory studies that included rodent collections from the southern tip of Sweden to above the Arctic Circle. We captured rodents throughout the country, attempted virus isolation from lung tissues, and tested sera for antibodies to Puumala virus. We also obtained human sera from many different parts of the country and assayed for anti-Puumala virus antibodies. The human serology found the highest prevalence rates and the highest incidence of infection in northern Sweden, while the rodent survey found the bank vole, *Myodes glareolus* (Previously known as *Cleithrionomys glareolus*) most frequently seropositive, with antibody-positive voles restricted to the northern two-thirds of the country and corresponding to the regions where the most human disease was seen. Through our studies, we were also able to isolate Puumala virus from the lung tissue of captured bank voles and propagate the virus in cell culture. Together with the field study results, we were able to conclude that the bank vole was the principal host of Puumala virus in Sweden and that the region near 64 degrees N was highly endemic for nephropathia epidemica in humans [18,19].

## 8. Hantaviruses in the Balkan Countries

Hemorrhagic fever with renal syndrome was also known to occur in the Balkan countries prior to the isolation of Hantaan virus by H.W. Lee. A mild form of HFRS similar to nephropatia epidemica of Scandinavia was known, but in addition, a more severe disease with significant mortality existed. We were able to initiate collaborations with colleagues in both Greece and what was then Yugoslavia with the goal of determining the hantaviruses present and to further refine our understanding of their natural history.

In Greece, we worked with a physician-scientist and professor of microbiology at the Aristotle University of Thessaloniki, “Tony” Antoniadis. Tony had collaborated with USAMRIID scientists in the past and was eager to further his research on the hantaviruses. Jointly, we worked to enhance his laboratory capabilities with shared reagents and exchange visits for training and technology transfer and to develop research topics. In the summer of 1983, a small outbreak of severe HFRS occurred in the mountainous region of northwestern Greece involving eight patients. Those infected were shepherds who had camped in mountain pastures while tending their sheep. All were hospitalized, three were severely ill, and one died [20]. The following spring, we conducted field studies to obtain serum samples from village residents where four cases resided and to attempt rodent collections in the mountain pastures where the shepherds had camped. We did serological tests on patients and village residents and found that all patients had anti-hantavirus antibodies, as did about 4% of the villagers. When we looked for neutralizing antibodies to Hantaan, Seoul, and Puumala viruses, we found the highest titers to Hantaan virus, although the antibody titers were generally less than those seen for known Hantaan-infected patients in Asia. These results suggested that a yet-to-be-recognized hantavirus may have been responsible for the more serious disease. The rodent collections found only *Apodemus flavicollis* rodents seropositive, and we were unable to isolate a virus from the tissues of the seropositive rodents. This and other studies in Greece suggested that severe HFRS was not due to either Puumala or Seoul virus but rather a novel hantavirus more closely related to Hantaan virus of Asia. Further, it suggested that another rodent species, *Apodemus flavicollis*, may have been the host of this suspected new virus [21].

Prior to the civil war in Yugoslavia, there was an initiative to foster greater collaborations between the US and Yugoslavian militaries, and, as a result, our program on the hantaviruses was selected by the US Army as a topic for joint research. Yugoslav scientists had a long history of work on viral diseases, including on smallpox virus, and at this time in the 1980s, Ana Gligic was one of the premier leaders in virology in Yugoslavia. Ana had collaborated with Carlton Gajdusek at NIH, had strong ties to the Yugoslav military academy in Belgrade, and had established collaborations with promising young scientists in Ljubljana and Sarajevo [22]. I first met Ana at a small international meeting at Smolenice Castle outside of Bratislava in what was then Czechoslovakia. This unique meeting that brought together scientists from both sides of the Iron Curtain was remarkable in demonstrating that science crosses political boundaries and that the common desire to better understand the natural world around us is universal. The meeting set the stage for collaborations with Ana and her team to build on our mutual interests in the hantaviruses and establish collaborative studies and exchanges.

In the spring of 1989, we set out to replicate the ribavirin antiviral drug trial of China in Sarajevo in hopes of demonstrating the drug’s efficacy in a second population and against the severe form of HFRS found there. We identified a talented team of young physicians just graduating from the medical school in Sarajevo, led by Alemka Markotic, who had been number one in her class (see Figure 1). As we had done in China, we established a modern diagnostic lab at the medical school in Sarajevo to confirm the diagnosis of patients to be enrolled in the study. We were about to begin the study when the war broke out in Yugoslavia, dashing all hopes of doing the study and throwing the local team into a war footing that placed each of them in incredible danger as they attempted to care for those injured in the war. Indeed, at least one of our Sarajevo team members was wounded during this terrible time.

By the time the war started, I had retired from USAMRIID and taken a position with the Centers for Disease Control and Prevention (CDC), with my first assignment being seconded to the World Health Organization (WHO) in Geneva as an expert on arboviruses and hemorrhagic fevers. While at WHO, I was able to communicate with Alemka in Sarajevo as the war unfolded. Realizing the dire conditions and the dangers she faced, I was able to arrange for her transport to Geneva on a UN flight, and once safely in Geneva, she was able to help as we attempted to assess health risks in the region. Even many years later, she still thanks me on the anniversary of her evacuation to safety.

One of the more significant accomplishments made during this decade was the isolation of Dobrava virus by Tatjana Avsic-Zupanc and her colleagues, including scientists at USAMRIID [23]. They successfully isolated the new virus from the lungs of *Apodemus flavicollis* and were able to demonstrate serologically and through genomic analysis that the new virus was related to, but distinct from, all known hantaviruses. (At about the same time, a second team led by Ana Gligic isolated an identical virus that they named Belgrade virus; however, over time, the two have been shown to be the same, and the name Dobrava virus has persisted.) Dobrava virus was shown to be associated with severe HFRS and is now known in many countries of Europe, fortunately at a low prevalence of human infection in most places.

Our international collaborations yielded several very significant benefits. Using a National Research Council fellowship program, we were able to host talented young scientists to work on the hantaviruses at USAMRIID. This offered a path that allowed scientists from Asia, Europe, and other regions, as well as from the USA, to participate in extensive training and joint research that benefited everyone and truly advanced knowledge of the hantaviruses. These collaborations were critical to the development of the hantavirus community to which we all belong.

After a decade of truly remarkable hantavirus discoveries, the USAMRIID “dream team” began to unravel. The greatest loss was the untimely death of Joel Dalrymple in 1992. Joel had been the driving force of the program and was very close to Ho Wang Lee. They communicated often and were more than just scientific colleagues; they were true friends. Around the time of Joel’s passing, many of the senior officer-scientists faced mandatory military retirement and were faced with few options to continue at USAMRIID. Some moved to the CDC, where options to continue work on emerging infectious diseases were greater, resulting in much of the knowledge base and critical reagents that had been painstakingly developed at USAMRIID moving with these experienced leaders. This influx of talent and skills helped rapidly build the hantavirus program as an important element of the CDC’s fledgling program in emerging infectious diseases. Finally, funding for USAMRIID was evolving, making it harder to sustain the robust hantavirus research program that had been so productive over the past decade. While critical hantavirus investigations continued at USAMRIID, and many important discoveries are still being made by the excellent staff even today, the net result was a clear erosion of what had been one of the premier centers of excellence in hantavirology anywhere in the world.

When a completely new hantavirus causing hantavirus pulmonary syndrome (HPS) emerged in the Western US in 1993, the CDC was well positioned to respond, thanks in large part to the decade of productive research facilitated by USAMRIID and supported by the Department of Defense. As the HPS story evolved, “We” were able to build on our past international collaborations pioneered at USAMRIID as well as those already in place at the CDC to help address HPS in Central and South America. “We” quickly discovered novel hantaviruses that had caused serious and often fatal human infections in many locations, but whose cause had never been known. In addition, “We” discovered many new rodent hosts of these novel viruses. Today, our network continues to grow internationally as we discover yet more hantaviruses involving an ever-increasing array of animal hosts and perhaps still others causing more human diseases that have yet to be recognized.

## 9. Final Thoughts

Since the original isolation of Hantaan virus by Ho Wang Lee, “We” have all contributed to and indeed benefitted from the enormous power of collaborations. It’s better to collaborate than to compete: everyone wins in a collaboration. Through mutual respect and shared vision, “We” have accomplished much and seen the tremendous rewards of working together. “We” have built a knowledge base on an important group of viruses that has been the scourge of society for ages. “We” have worked together to make fundamental discoveries and to develop interventions to mitigate the impact of infections, and “We” have shared our findings with one another through informal conversations, joint meetings, and scientific publications. Our reputations as trusted collaborators will facilitate our continued success; our networks of collaborators are our most valuable assets—we must treasure them. “We” have invested in training future generations of hantavirologists, and “We” have worked to ensure their success. The investments made in nurturing junior colleagues will pay a lifetime of dividends.

“The Power of We” is all of us, working together, continuing the vision of Ho Wang Lee and Joel Dalrymple (see Figure 2).

## Figures and Tables

**Figure 1 viruses-15-00921-f001:**
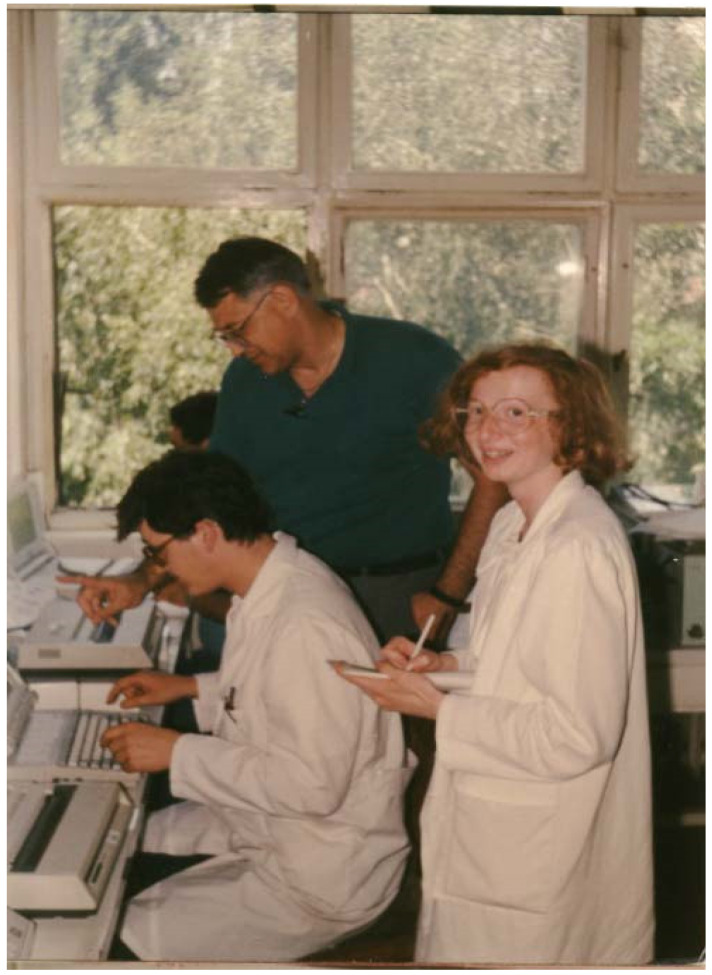
Dado Sarvecic, Tom Ksiazek, and Alemka Markotic in Sarajevo, 1989, setting up hantavirus diagnostic testing for planned ribavirin efficacy trial.

**Figure 2 viruses-15-00921-f002:**
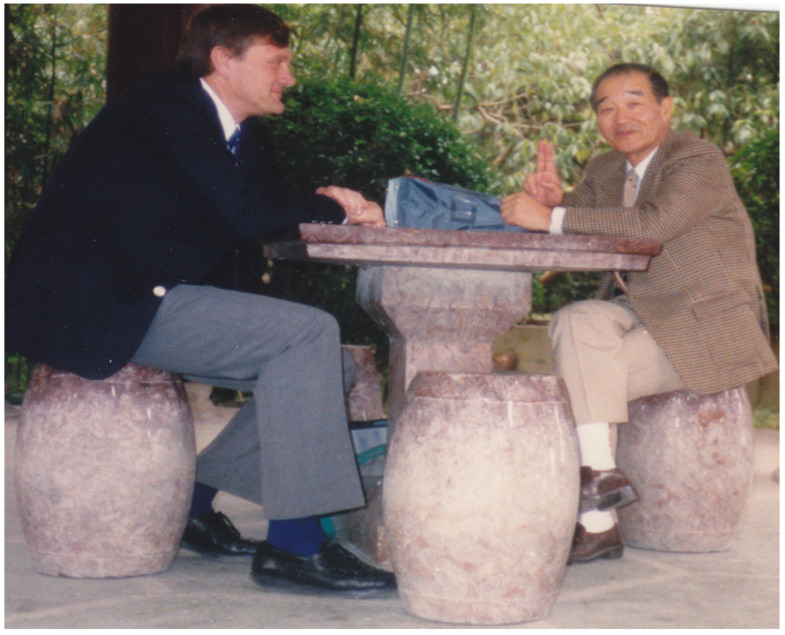
Joel Dalrymple and Ho Wang Lee. Their mutual respect and admiration created a foundation of collaboration that has benefited the entire field of hantavirology.

**Table 1 viruses-15-00921-t001:** Sera positive for IgM-specific anti-Hantaan virus antibodies by disease day from Korean hemorrhagic fever patients collected during the Korean conflict. Disease day is the number of days post onset of clinical symptoms. (Reproduced from [2]).

Disease Day	Total Positive/Total Tested (%)
1	3/5 (60)
2	25/41 (61)
3	47/62 (76)
4	35/40 (88)
5	20/21 (95)
6	14/15 (93)
7	9/9 (100)
8	9/9 (100)
9	1/1 (100)
10	6/6 (100)
11–20	100/100 (100)
21–30	149/149 (100)
31–40	44/44 (100)
41–50	42/42 (100)
>50	30/30 (100)

**Table 2 viruses-15-00921-t002:** Results of immunofluorescent antibody (IFA) tests for Hantaan antibodies in urban rats in the United States. (Reproduced from [4]).

Locations and Dates	No. Pos/Tested (% Pos.)
**Philadelphia, PA: November–December 1981**	
*Container-wharf*	*0/53*
*Naval shipyard*	*2/20 (10%)*
*Incinerator*	*1/24 (4%)*
*Girard Point granary*	*4/13 (31%)*
**Total**	**7/110 (6%)**
*Repeat at granary: March 1982*	*8/12 (67%)*
**Houston, TX: January 1982**	
*Seamen’s Center*	0/9
*Granary no. 1*	0/31
*Wharf for granary no. 1*	5/8 (63%)
*Granary no. 2*	0/7
*Granary no. 3*	1/17 (6%)
*Granary no. 4*	0/19
*Galveston granary*	0/18
**Total**	**6/109 (5%)**
*Repeat at wharf for Granary no. 1: February 1982*	3/6 (50%)
**Various California ports: February 1982**	
*Sacramento*	*0/18*
*Oakland*	*0/5*
*San Francisco*	*0/28*
*Los Angeles*	*0/5*
**Total**	**0/28**

**Table 3 viruses-15-00921-t003:** IFA antibodies to Hantaan virus among *Rattus rattus* and *Rattus norvegicus* captured in South American cities, September 1982 to March 1983. (Reproduced from [15]).

Location	Date Captured	IFA Pos/Total (%)
**Brazil**		
Belem	Feb 1983	30/54 (56%)
Sao Paulo	Nov-Dec 1982	5/35 (14%)
Recife-Olinda	Sept-Nov 1982	2/36 (6%)
**Argentina**		
Buenos Aires	Feb-Mar 1983	11/101 (11%)

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
