# Peer review of "The Power of We"

_viruses, 2023, doi:10.3390/v15040921_

Round 1
Reviewer 1 Report
I have marked a number of suggested changes (all minor) throughout the manuscript. Also I commented on some problems with the citations and references.

Author Response
Please see attached for responses to specific questions. The revised manuscript was attached to the earlier reply.
Please let me know if you failed to receive or have additional questions.
Please extend my thanks to the reviewers for their helpful comments and suggestions.

Reviewer 2 Report
This was an enjoyable personal account of the research teams involved in early hantavirus discovery written by an accomplished researcher. The supportive, collaborative sentiments voiced in this perspective were welcome and inspirational. Overall, this is a positive and enjoyable perspective.
I only have a few minor comments:
1. The figures taken from previous publications are all blurry. I understand the publications are likely difficult to access and the images there may have been of lower quality than what is used today. However, as presented the figures are impossible to read.
2. On line 70, as written gives the impression that Drs. Ksaizek and Meegan invented the ELISA. Younger readers may not realize that the assay was invented independently by Drs. Engvall and Perlman over 15yrs prior. Should be rewritten.
3. Lines 113-116 could benefit from citations
Author Response
- I realize the poor quality of the figures and unfortunately I do not have access to the original data to reproduce them; consequently, I have deleted all figures. Let me know if you have suggestions on how to salvage them.
- Line 70 comment has been reworded to indicate that the EIA was adapted for use on the hantaviruses.
- A reference for lines 113-117 has been added as requested.
MDPI review comments: I agree that the manuscript is better referenced as a review article.
References have been modified as requested.
Please see attachment with changes as noted.
